# Towards Detecting Insider Trading in Prediction Markets: Event-Aligned Multi-Scale Anomaly Detection

Youwen Wu
*Department of Computer Science and Technology*
*Tsinghua University*

Jolin Zhu
*Department of Computer Science and Technology*
*Tsinghua University*

Claire Insixiengmay
*School of Life Sciences*
*Tsinghua University*

## I. Introduction

A prediction market is a market where users can trade contracts whose payoff depend on the outcomes of future events. In other words, they "trade on their beliefs." For instance, a binary prediction market might ask whether the U.S. President will make a statement about the economy by a certain fixed date. An $n$-ary market might assign probabilities to $n$ different outcomes. For instance, a market might ask which politician is most likely to win the U.S. Presidential Election. The price of a "share" of an outcome is directly tied to the market's forecasted probability—in Polymarket (USA), an event with probability $p \in [0, 1]$ has share price $p$ in U.S. Dollars [1]. The liquidity and pricing of shares in a market is governed by algorithms such as Logarithmic Market Scoring Rule (LMSR) [2].

Detecting anomalous trading activity is highly important as demonstrated by the recent concerns over insider trading sparked by the arrest of a U.S. soldier who allegedly bet on the arrest of Nicolás Maduro, President of Venezuela, with insider information on the military operations leading to Maduro's capture [3].

We propose a lightweight, context-aware anomaly detection architecture for prediction markets that combines multi-scale time-series analysis with event alignment. Historical market prices are transformed into logit probability space and processed through two complementary branches: a low-frequency trend detector for gradual repricing and a high-frequency shock detector for abrupt jumps. Their outputs are fused into a market anomaly score, which is then combined with an event-context layer based on manually curated public event timestamps and explanation checks. The final output is a ranked set of anomalous market windows labeled as explained, partially explained, or unexplained, where unexplained pre-event movements are treated as candidate cases for further investigation into possible insider trading or other abnormal informed trading behavior.

### A. Gaps Identified

Previous time-series and deep learning models can only identify statistical anomalies in data but are fundamentally context-blind. They are unable to differentiate these anomalies and what the motivating cause behind them could have been.

There is limited integration of external information in these previous frameworks leading to lack of distinction between unexplained anomalies and those that could be explained by public information as well as lack of exploration in the temporal relationship between belief shifts, information release, and the anomaly. Additionally, previous works on sentiment analysis for stock market research lacks more exploration regarding more sophisticated analysis, such as topic modeling to gain better insight on how sentiment analysis can aid stock prediction [4].

### B. Novelty / Contribution

Models in the past primarily focused on improving anomaly detection performance, but remains solely data-driven without integration of external market context. Because of this they cannot distinguish past statistical irregularities as being reflected by public information or unexplained behavior. Our proposed architecture aims to add background and explanation behind an anomaly by taking external events into consideration. Going beyond just detection, by integrating external event signals, our proposed framework aims to enhance adaptability to real-world market conditions and provide interpretable insights behind abnormal market behavior.

## II. Motivation

Prediction markets are attractive targets for anomaly detection because their prices function as real-time, market-aggregated forecasts of uncertain events, making unusual price movements potentially informative signals about new information, manipulation, or informed trading. Prior work on prediction markets shows that market prices can aggregate dispersed information into accurate forecasts, but this same sensitivity means that abrupt or unexplained movements may deserve closer scrutiny. At the same time, financial time series are nonlinear, non-stationary, and multi-scale, so simple threshold-based jump detection is likely to confuse ordinary information updates with genuinely anomalous behavior; broader time-series anomaly detection literature emphasizes the need to model temporal context rather than isolated observations.

Our architecture is motivated by recent financial anomaly-detection work such as WaveLST-Trans, which separates low-frequency trend behavior from high-frequency fluctua-

tions before fusing them into an anomaly score. The paper argues that wavelet-based multi-scale decomposition helps distinguish long-term market trends from short-term abnormal shocks, while LSTM and Transformer components capture complementary temporal patterns. However, prediction markets require an additional contextual layer: a large price movement after public news may be normal, while a smaller movement before a public announcement may be more suspicious. Therefore, we adapt the multi-scale anomaly-detection idea into a lighter, feasible architecture that combines price-based anomaly scores with event-alignment and public-explanation checks.

The motivation for the proposed system is thus not necessarily to conclusively prove insider trading, but to create a disciplined triage mechanism for identifying candidate windows that merit further investigation. This is consistent with changepoint and modern unsupervised anomaly-detection approaches, which focus on detecting abrupt distributional shifts or unusual temporal associations rather than directly assigning legal or causal labels. By combining multi-scale market behavior with event timing, the architecture aims to reduce false positives from ordinary news-driven repricing while surfacing unexplained pre-event movements as higher-priority investigative leads.

## III. Background

Prediction markets are platforms where participants can buy and sell contracts based on future events and in this way can be used to aggregate private information. Market makers and liquidity mechanisms, such as the Logarithmic Market Scoring Rule, are used to maintain tradable prices even when natural liquidity is limited [2]. By trading with payoffs that depend on defined outcomes, these trades directly move prices, making the prices interpretable as the probability the market assigns to an event. In this way prediction markets become a collective forecast [5]. However, this also means price movements are susceptible to heterogeneous information among traders.

For our task of anomaly detection, to avoid previous limitations from other models, we adopt the WaveLST-Trans model. The model consists of a wavelet transform preprocessing module, an LSTM long-term dependency modeling module, a Transformer short-term fluctuation capturing module, and a feature fusion layer [6]. In this way we can ensure that anomaly detection is multi-scale, and account for different movements in the prediction-market like abrupt jumps or slow drifts by having a branch for both low-frequency repricing and another for high-frequency shocks.

After detection, we aim to extend this structure with an event-alignment module. This is the main domain-specific contribution. In the early history of this anomaly detection, researchers relied on statistical assumptions to detect outliers as anomalies and then moved on to econometric methods like event studies to analyze events. Recent machine learning and deep learning approaches add context and multimodal approaches by integrating external signals; however they still lack residual analysis and do not distinguish between anomalies explained by public information and those that are not.

## IV. Related Works / Existing Methods

a) *Classical Statistical Models:* Traditional methods in the field of time series anomaly detection on the market have mostly been reliant on statistical models such as the Autoregressive Integrated Moving Average (ARIMA) which detects anomalies by modeling the linear dependencies of data. Thus, it is limited when it comes to its adaptability to nonlinear and abrupt changes [7]. Others, like the Generalized Autoregressive Conditional Heteroskedasticity (GARCH) model and its variants (such as EGARCH and TGARCH) focus on volatility modeling in financial markets and effectively describe the heteroskedasticity of financial time series, but still struggle suffer in identifying sudden anomalies due to delays [8].

b) *Deep Learning Models:* As of late, deep learning has also been widely applied in financial time series anomaly detection such as Long Short-Term Memory (LSTM) networks that are able to model long-term dependencies but are vulnerable to the vanishing gradient problem and limited in capturing abrupt, short-term anomalies [9]. Gated Recurrent Units (GRU), a simplified version of LSTM, decrease computational complexity but still struggle with modeling local mutations in time series [9]. Convolutional Neural Networks (CNNs) have also been applied to financial time series feature extraction despite their use for computer vision tasks, but since they rely on fixed-size receptive fields, they perform poorly with non-stationary time series data due to their difficulty in capturing long-range dependencies [10]. Hybrid models combining LSTM with CNN optimize time series modeling efficiency but still face shortcomings like high computational complexity and limited generalization capabilities [11].

c) *Anomaly Detection:* Other anomaly detection methods such as Isolation Forest, an unsupervised method, identifies outliers by random partitioning of data points, making it suitable for high-dimensional data but does not account for temporal dependencies [12]. The Local Outlier Factor (LOF) calculates local abnormality based on data density distributions and is effective for detecting certain patterns of anomalies, but is highly sensitive to noisy data [13]. Support Vector Machines (SVM) and its extended versions, such as One-Class SVM, distinguish between normal and anomalous data by constructing hyperplanes in high-dimensional space and are suitable for static data, but overall cannot handle the dynamic nature of the market [13]. Newly researched methods like the WaveLST-Trans model addresses these shortcomings by combining wavelet transform (WT), LSTM and Transformer to adapt to the non-linear, multi-scale characteristics of financial time series by creating a complementary framework for modeling both long and short-term dependencies. However, it lacks the integration of external market events and therefore cannot distinguish between anomalies driven by public news or those reflecting unexplained behavior [6].

d) *Traditional Hybrid Sentiment Analysis:* Compared to other research done on sentiment analysis for stock markets, prediction markets require a more nuanced NLP pipeline, as specific event outcomes are directly traded on, rather than the abstract concept of a company's valuation [4].

## V. Challenges

### A. Data Challenges

Prediction market data is sparse for low-liquidity events, making statistical anomaly detection noisy. Unlike large financial markets, a prediction market may not have long-term observations and can also have missing intervals or sudden liquidity changes. Additionally, public event data such as news, statements is unstructured, multi-lingual, and lacks standardized timestamps, complicating alignment with market ticks.

### B. Modeling Challenges

Disentangling information-driven price movements from genuinely anomalous trading requires causal inference under severe confounders—public and private information often arrive simultaneously. The model must also balance sensitivity, by catching real insider trading, with specificity, avoiding false alarms from legitimate rapid repricing.

### C. Validation Challenges

With respect to verifying against publicly available real-world datasets, there exist few publicly known and verified insider trading cases. Due to the sparsity of true insider trading labels, we cannot prove insider trading and success is only measured by if event alignment improves interpretability rather than being a legal classifier. There is also a lack of complete manual event labels and relevant event news could be missed or hidden due to it being hard to observe.

## VI. Objectives

The goal of the project is to design and evaluate a context-aware anomaly detection framework for prediction market time series. The system should be able to identify statistically unusual market movements and then interpret them alongside public information release timing. The intended output would be a ranked list of anomalous windows labeled as explained, partially explained, or unexplained.

Beyond anomaly detection, we aim to compare price-only anomaly detection with event-aligned anomaly detection by testing whether adding event context can reduce false positive caused by ordinary news reactions.

## VII. Proposed Methodology

### A. Overall Pipeline / Architecture

We begin with Polymarket data, a curated public event table, and a pre-defined set of official sources, news sources, and public fora to scrape information from. For instance, official sources may include official Twitter accounts or press releases. Public fora include Twitter, Bluesky, and other social media where users engage in discussion. Additionally, we incorporate future scheduled announcements in advance, from the date their release schedules are known.

There are two primary components that first operate independently on the Polymarket data and the contextual information. For the former, we implement an adapted, sim-

plified version of the financial time series anomaly detection technique WaveletLST-Trans detailed in [6]. For the latter, we implement a sophisticated NLP pipeline that crucially incorporates *understanding* of the contract itself. For each contract, the system collects public text related to the contract from the sources described previously. Each collected artifact is tagged with timestamp (of creation). Finally, a fusion step triages the anomalies with their corresponding explainability, classifying each anomaly as "explained," "partially explained," or "unexplained" (i.e. actually anomalous).

*a) NLP Preprocessing:* In our context gathering pipeline, a text preprocessing system first performs deduplication, filtering, timestamp normalization, while the Polymarket contracts being investigated are semantically analyzed and evaluated for relevance to the processed text. The final relevant text artifacts obtained are called *events*.

*b) NLP Semantic analysis:* Each event is then characterized and assigned the corresponding outcome it implies in the market. For example, in a Yes / No binary outcome market, a relevant piece of text will be assigned a score in the direction of Yes / No. Events are also assigned novelty scores, based on whether they reiterate previously known information, as well as credibility scores based on source. News articles are more credible than public Twitter sentiment, for instance, while official statements have highest credibility.

We refer the reader to Fig. 2 in the Appendix for the full, detailed diagram of our NLP pipeline.

*c) Anomaly Detection:* We implement an anomaly detection system based on the WaveLST-Trans architecture proposed in [6], adapted from ordinary financial price series to prediction market probability series. The central motivation for using this architecture is that prediction market prices contain both slow-moving probability updates and abrupt jumps. A market may gradually drift as public sentiment or evidence accumulates, but it may also move sharply around a breaking news event, official announcement, or possible information leak. A single time-series model may fail to represent both patterns well. WaveLST-Trans addresses this problem by explicitly decomposing the input series into low-frequency and high-frequency components, modeling the former with an LSTM and the latter with a Transformer, then fusing the resulting representations into an anomaly score. The original paper describes this design as a wavelet preprocessing module followed by an LSTM branch for long-term dependencies, a Transformer branch for short-term fluctuations, a feature fusion layer, and a final anomaly detection layer.

A diagram of the architecture is in the Appendix at Fig. 3.

In our setting, the raw input is not a stock or cryptocurrency price, but a Polymarket contract price. Since a binary prediction market price can be interpreted as an approximate probability, we first transform each price series into logit space. We then compute first differences or returns in logit space, producing a time series of belief updates. Where available, we also include auxiliary market features such as trading volume, liquidity, bid-ask spread, or market depth. These features are normalized and segmented into fixed-length sliding windows, following the sliding-window preprocessing strategy described in WaveLST-Trans.

After preprocessing, we apply a discrete wavelet transform to each market window. The purpose of this step is to separate slow-moving trend behavior from short-term fluctuations. The low-frequency component captures gradual repricing, such as a steady increase in the probability of an event as public evidence accumulates. The high-frequency component captures abrupt local deviations, such as a sudden jump in price or volume shortly before an announcement. This mirrors the original WaveLST-Trans design, where wavelet decomposition produces low-frequency components for long-term trend modeling and high-frequency components for short-term fluctuation modeling. The original paper specifically uses wavelet decomposition to improve multi-scale feature representation before feeding the separated components into different neural modules.

The low-frequency component is passed into an LSTM branch. This branch models longer-term dependencies in the prediction market's price trajectory. In our context, the LSTM is intended to learn patterns such as gradual belief revision, sustained drift, and normal pre-resolution convergence. This is important because not all large movements are suspicious: a market may move gradually and legitimately as public information accumulates. The LSTM branch therefore helps establish a learned representation of the market's expected longer-run trajectory, against which unusual deviations can be compared.

The high-frequency component is passed into a Transformer branch. This branch is designed to detect abrupt local patterns, including sudden jumps, volatility bursts, and short-lived trading shocks. The Transformer is appropriate here because its self-attention mechanism can compare different points within a window and identify unusual local relationships. In the original WaveLST-Trans architecture, the Transformer receives the high-frequency component extracted by wavelet transform and applies multi-head self-attention with positional encoding to model short-term fluctuations. We retain this split because our task is especially sensitive to sudden pre-event movements, which may be exactly the kind of high-frequency anomaly that a trend model alone would smooth away.

The outputs of the LSTM and Transformer branches are then passed into a feature fusion layer. This fusion layer combines the low-frequency trend representation with the high-frequency shock representation. In the simplest implementation, the two vectors can be concatenated and passed through a fully connected layer. A more flexible implementation can use learned adaptive weights, allowing the model to rely more heavily on the high-frequency branch when detecting sudden jumps and more heavily on the low-frequency branch when identifying abnormal drift. The fused representation is then mapped to a market anomaly score for each time window.

The final output of this stage is not yet an insider-trading label. It is only a time-series anomaly score. A high score indicates that the prediction market moved unusually relative to its learned multi-scale price dynamics. The model may flag several types of behavior: abrupt jumps, sustained abnormal drift, volatility bursts, or unusual changes in volume/liquidity if those features are included. These flagged windows are

then passed to the context-aware fusion stage, where they are compared against the NLP-derived public event timeline.

This separation is important. The anomaly detector answers the question: "Did the market move in a statistically unusual way?"

The NLP context module answers the question: "Was there public information that plausibly explains this movement?"

Only after these two signals are fused do we classify the anomaly as explained, partially explained, or unexplained. Thus, the WaveLST-Trans component provides the market-side evidence, while the NLP pipeline provides the public-information explanation layer. This design prevents the system from equating all large price movements with suspicious behavior. A large post-announcement move may be statistically anomalous but contextually explained, while a smaller pre-announcement move may be more interesting if it lacks a public explanation.

In summary, our anomaly detection module adapts WaveLST-Trans to prediction markets by replacing ordinary asset prices with logit-transformed prediction market probabilities, preserving the wavelet-based separation between low-frequency trends and high-frequency shocks, and using the LSTM/Transformer split to generate a market anomaly score. This score is then fused with event-context signals from the NLP pipeline to produce the final triage classification.

A detailed mathematical description of WaveLST-Trans is in the Appendix at Section XI.A.

d) *Context-Aware Fusion and Final Labeling:*

The fusion step combines the market-side anomaly score produced by the WaveLST-Trans module with the public-information signals produced by the NLP pipeline. The purpose of this step is to distinguish between price movements that are merely statistically unusual and price movements that are unusual *and* difficult to explain using publicly available information. This distinction is central to our project: a prediction market may move sharply after a major public announcement, but such a movement should not be treated as suspicious simply because it is large. Conversely, a smaller movement before a public announcement may be more important if it occurs without any clear public explanation.

The input to the fusion module is a set of flagged market windows. Each window has a market anomaly score $M_t$, which measures how unusual the price movement is relative to the model's learned time-series dynamics. For each flagged window, the NLP pipeline retrieves nearby relevant public events and assigns each event several scores: relevance to the contract, directional effect on the market outcome, novelty, source credibility, and timing relative to the price movement. These signals are then used to compute a public explanation score.

For a flagged market window at time $t$, we define the public explanation score $E_t$ as a function of the strongest nearby public event:

$$E_t = f(\text{relevance}_t, \text{direction}_t, \text{novelty}_t, \text{credibility}_t, \text{timing}_t)$$

The fusion module also computes a pre-event timing score $P_t$, which is high when the market movement occurs before the first identified public event that plausibly explains it.

The final context-aware suspiciousness score is:

$$S_t = \alpha M_t + \beta P_t - \gamma E_t \tag{2}$$

Here, $M_t$ is the market anomaly score, $P_t$ is the pre-event timing score, and $E_t$ is the public explanation score.

We also define a liquidity confidence score:

$$Q_t = g(\text{volume}_t, \text{spread}_t, \text{depth}_t, \text{trade\_count}_t) \tag{3}$$

The final label is assigned using rule-guided logic:

$$\text{label}_t = \begin{cases} \text{Explained} \\ M_t > \tau_M \text{ and } E_t > \tau_E \text{ and } P_t \leq \tau_P \\ \text{Unexplained} \\ M_t > \tau_M \text{ and } E_t \leq \tau_E \text{ and } P_t > \tau_P \\ \text{Low-liquidity artifact} \\ M_t > \tau_M \text{ and } Q_t \leq \tau_Q \\ \text{Partially explained} \\ M_t > \tau_M \text{ and otherwise} \end{cases} \tag{4}$$

## VIII. Dataset

The dataset consists of historical price time-series from a set of prediction markets on Polymarket. This project will aim to select markets with sufficient trading activity and a clear connection to public events. Each record contains a timestamp and market price as well as possible additional features like volume, liquidity, or spread information. In addition to the Polymarket data, we will also use generative tools (e.g. LLMs) to create synthetic data for our evaluation set. This project will also include a manually curated event table with each event including a timestamp, event description, source type, and relevance label. Preprocessing will include timestamp standardization, price cleaning, logit transformation, return calculation, rolling-window feature construction, and alignment between market data and event records.

## IX. Project Schedule

See Table 1 below.

## X. Progress So Far

We have established the general scope of the project and narrowed down our objective from an insider trading detection system to a anomaly-triage architecture. We have completed a review of related works regarding anomaly detection for financial time-series and past multi-scale architectures used for the task to identify architectural references that we would want to incorporate in our model, such as the WaveLST-Trans. We have also done research in previous relevant work on prediction markets. From there, we have also narrowed down the structure to have a more feasible framework rather than the entire wavelet-LSTM-Transformer implementation. The overall pipeline architecture has been mapped out and the necessary datasets have been obtained.

As for ongoing work, we are experimenting with the WaveLST-Trans model and adjustments to either simplify it for our use cases or specialize it for the particular domain of prediction markets. We are also finalizing details for the synthetic benchmark so that we can evaluate our dataset. Once the implementation details are finalized, full work on the described systems will proceed.

## XI. Appendix

### A. Detailed description of WaveLST-Trans

The following is based primarily on work in [6].

The input to the anomaly detector is a sliding window of Polymarket observations. For a binary contract with price $p_t$, we first transform the price into logit space:

$$z_t = \log\left(\frac{p_t}{1 - p_t}\right). \tag{5}$$

This transformation is used because binary prediction market prices are bounded between 0 and 1, while changes in market belief are more naturally modeled on an unbounded log-odds scale. We then compute first differences,

$$r_t = z_t - z_{t-1}, \tag{6}$$

TABLE I
Project Schedule

| Week | Dates | Phase | Tasks | Lead / Support |
|------|-------|-------|-------|----------------|
| 1 | 4/30–5/10 | Literature and data selection | Finalize scope, review prediction-market and anomaly-detection literature, identify accessible data sources, and define event categories. | Youwen: architecture and literature; Jolin: data source review; Claire: event-context design |
| 2 | 5/10–5/22 | Preprocessing and baseline detectors | Collect initial market price data, clean timestamps, apply logit transformation, compute returns and rolling-window features. | Jolin: data preprocessing; Youwen: detector design; Claire: documentation |
| 3 | 5/22–5/31 | Event alignment and fusion | Construct event table, align detected anomalies with public events, implement simple scoring rules for pre-event, on-event, and post-event windows, and generate ranked anomaly outputs. | Claire: event table; Youwen: fusion scoring; Jolin: analysis scripts |
| 4 | 5/31- 6/7 | Evaluation and final report | Compare price-only and context-aware outputs, prepare case studies, analyze limitations, write final report, and prepare final presentation. | Youwen: final methodology and writing; Jolin: results and figures; Claire: motivation, limitations, and presentation |

which represent market belief updates. Where available, we concatenate additional market features such as volume, liquidity, bid-ask spread, and market depth. Thus each time step is represented as a feature vector

$$x_t = [z_t, r_t, \text{volume}_t, \text{liquidity}_t, \text{spread}_t, \text{depth}_t]. \quad (7)$$

Missing features are either omitted or imputed with explicit missingness indicators. The resulting sequence is normalized using statistics estimated on the training split and divided into fixed-length windows

$$X_t = \{x_{t-W+1}, ..., x_t\}. \quad (8)$$

Following the WaveLST-Trans design, each input window is passed through a discrete wavelet transform. We use the wavelet transform to decompose the sequence into low-frequency and high-frequency components:

$$X_t \rightarrow (X_t^L, X_t^H). \quad (9)$$

The low-frequency component $X_t^L$ captures gradual probability drift, long-term repricing, and normal convergence toward market resolution. The high-frequency component $X_t^H$ captures sudden jumps, local volatility bursts, and short-lived shocks. We adopt a Daubechies wavelet, such as db4, following the design choice in WaveLST-Trans, because it provides a compact representation of both smooth trend behavior and localized fluctuations. The decomposition level is treated as a hyperparameter, with a small value such as 2 or 3 preferred to avoid over-smoothing short-lived prediction market movements.

The low-frequency component is passed to an LSTM branch. The LSTM is responsible for modeling longer-term temporal dependencies in the market. In our setting, these dependencies include gradual belief revision, sustained pre-resolution drift, and repeated market-specific patterns in how probabilities adjust over time. Let the LSTM produce a hidden representation

$$h_t^L = \text{LSTM}(X_t^L). \quad (10)$$

The high-frequency component is passed to a Transformer encoder branch. The Transformer branch is responsible for modeling abrupt local patterns in the high-frequency signal. Since sudden jumps shortly before an announcement are especially important for our task, the self-attention mechanism is useful because it can compare different time steps within the same window and identify unusual local relationships. We apply positional encoding to retain temporal order and obtain

$$h_t^H = \text{Transformer}(X_t^H). \quad (11)$$

The two branch outputs are then combined in a feature fusion layer. In the baseline implementation, we concatenate the two representations and pass them through a fully connected layer:

$$h_t = \varphi\big(W_f[h_t^L; h_t^H] + b_f\big), \quad (12)$$

where $\varphi$ is a nonlinear activation function. A more flexible version may use learned gating weights,

$$h_t = \alpha_t h_t^L + (1 - \alpha_t) h_t^H, \quad (13)$$

where $\alpha_t$ is learned from the input window. This allows the model to rely more heavily on the Transformer branch

for abrupt jumps and more heavily on the LSTM branch for abnormal long-term drift.

Because confirmed anomaly labels are likely sparse or unavailable, the primary training objective is self-supervised. We train the model to predict the next market update and optionally reconstruct the input window. The prediction head outputs

$$\hat{r}_{t+1} = f_{\text{pred}(h_t)}, \quad (14)$$

and the reconstruction head outputs

$$\hat{X}_t = f_{\text{rec}(h_t)}. \quad (15)$$

The prediction loss is defined using Huber loss rather than ordinary squared error:

$$L_{\text{pred}} = \text{Huber}\big(r_{t+1} - \hat{r}_{t+1}\big). \quad (16)$$

Huber loss is preferred because financial and prediction market time series are heavy-tailed: large moves should influence training, but they should not dominate the entire objective. The reconstruction loss is defined as

$$L_{\text{rec}} = \|X_t - \hat{X}_t\|_2^2. \quad (17)$$

The combined self-supervised loss is

$$L = \lambda_{\text{pred}} L_{\text{pred}} + \lambda_{\text{rec}} L_{\text{rec}}. \quad (18)$$

If hand-labeled anomaly windows are available, we add an optional supervised classification head

$$\hat{y}_t = \sigma\big(f_{\text{cls}(h_t)}\big), \quad (19)$$

with a binary cross-entropy or focal loss term:

$$L_{\text{cls}} = \text{BCE}(y_t, \hat{y}_t). \quad (20)$$

Focal loss may be preferred when labeled anomalies are highly imbalanced relative to normal windows. The full semi-supervised objective is then

$$L = \lambda_{\text{pred}} L_{\text{pred}} + \lambda_{\text{rec}} L_{\text{rec}} + \lambda_{\text{cls}} L_{\text{cls}}. \quad (21)$$

In the expected project setting, however, the self-supervised objective is the primary design, while manually labeled anomalies are mainly used for qualitative validation rather than large-scale supervised training.

After training, the model assigns each window a market anomaly score. The score combines prediction error, reconstruction error, and, if available, the supervised anomaly probability:

$$M_t = \eta_{\text{pred}} |r_{t+1} - \hat{r}_{t+1}| + \eta_{\text{rec}} \|X_t - \hat{X}_t\|_2^2 + \eta_{\text{cls}} \hat{y}_t. (22)$$

The weights $\eta_{\text{pred}}$, $\eta_{\text{rec}}$, and $\eta_{\text{cls}}$ control the relative importance of short-term forecasting error, structural reconstruction error, and supervised anomaly probability. In the simplest implementation, we omit the supervised term and use only prediction and reconstruction errors. A high value of $M_t$ indicates that the observed market behavior is difficult for the learned multi-scale model to explain.

Thresholds are calibrated on a held-out validation set. Since prediction markets differ greatly in liquidity and volatility, we avoid using a single universal raw threshold across all markets. Instead, we use market-specific or market-type-specific quantile thresholds, such as the 95th or 99th percentile of validation anomaly scores. This produces a binary flag

$$a_t = 1[M_t > \tau], \qquad (23)$$

where $\tau$ is the calibrated threshold. We may also retain the continuous score $M_t$ for ranking candidate anomalies rather than forcing a hard classification.

The output of the anomaly detection module is therefore a ranked set of anomalous market windows, each with a timestamp, market identifier, anomaly score, and supporting diagnostic quantities such as prediction error, reconstruction error, high-frequency wavelet energy, and volume abnormality. These windows are not treated as evidence of insider trading by themselves. Instead, they are passed to the context-aware fusion stage, where they are compared against the NLP-derived public event timeline.

This design deliberately separates statistical unusualness from contextual suspiciousness. The WaveLST-Trans module detects abnormal market dynamics, while the NLP module evaluates whether those dynamics are explained by public information. A large post-announcement movement may have a high market anomaly score but a low final suspiciousness score because it is contextually explained. Conversely, a smaller pre-announcement movement may become more important if it has a high anomaly score and no corresponding public explanation.

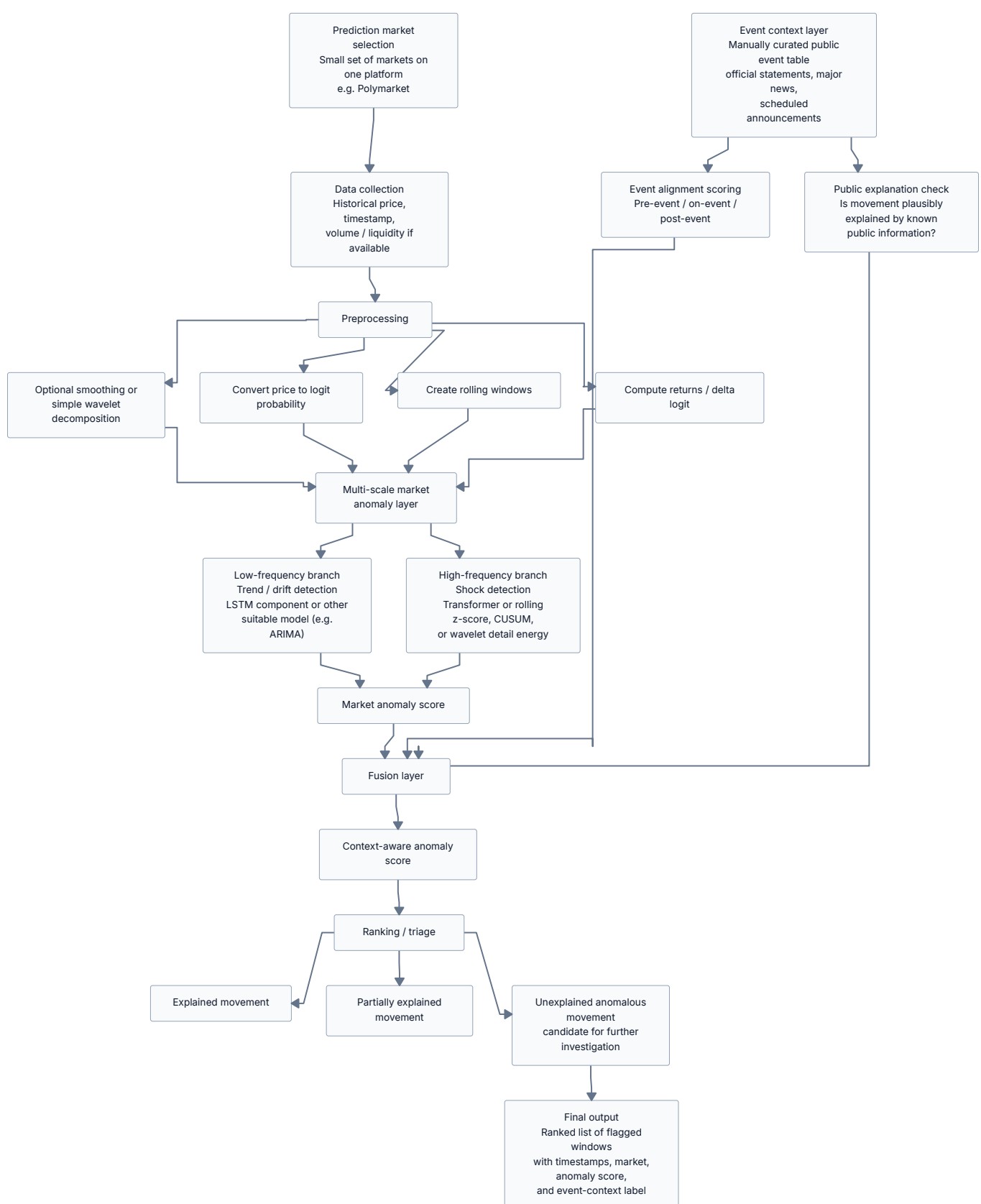

Fig. 1. Full detection architecture

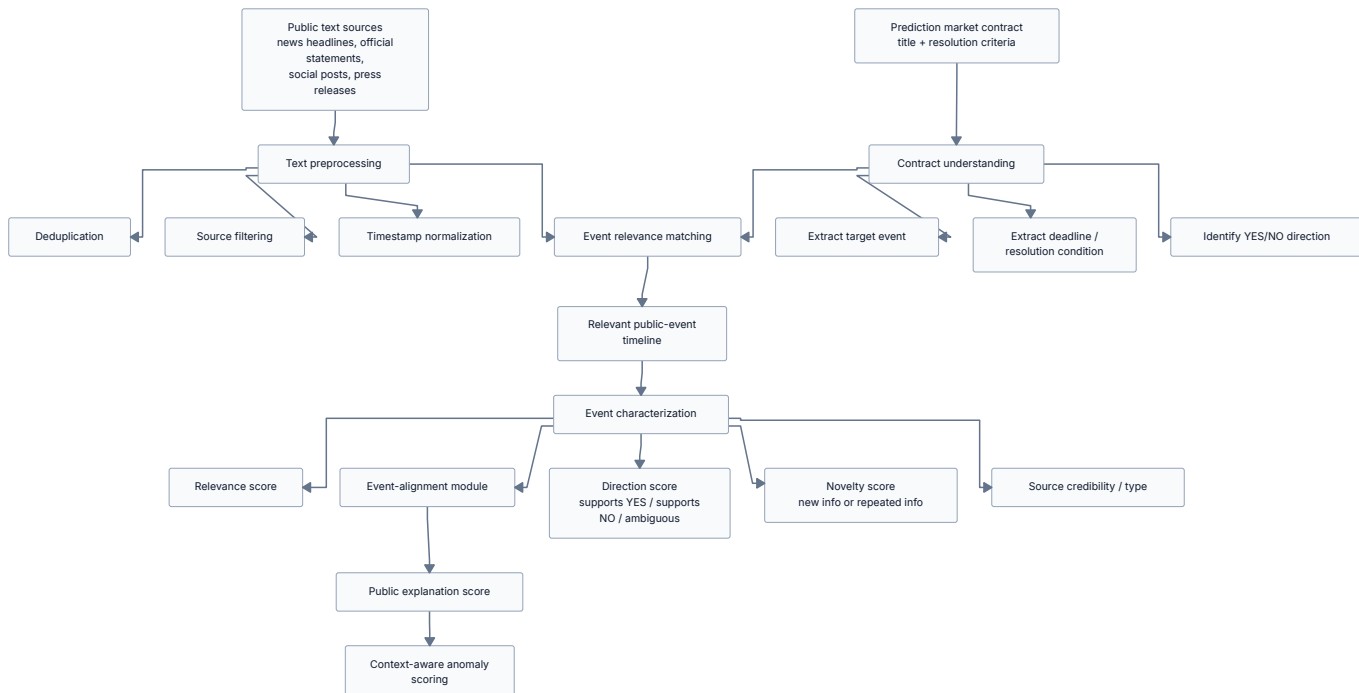

Fig. 2. NLP processing pipeline

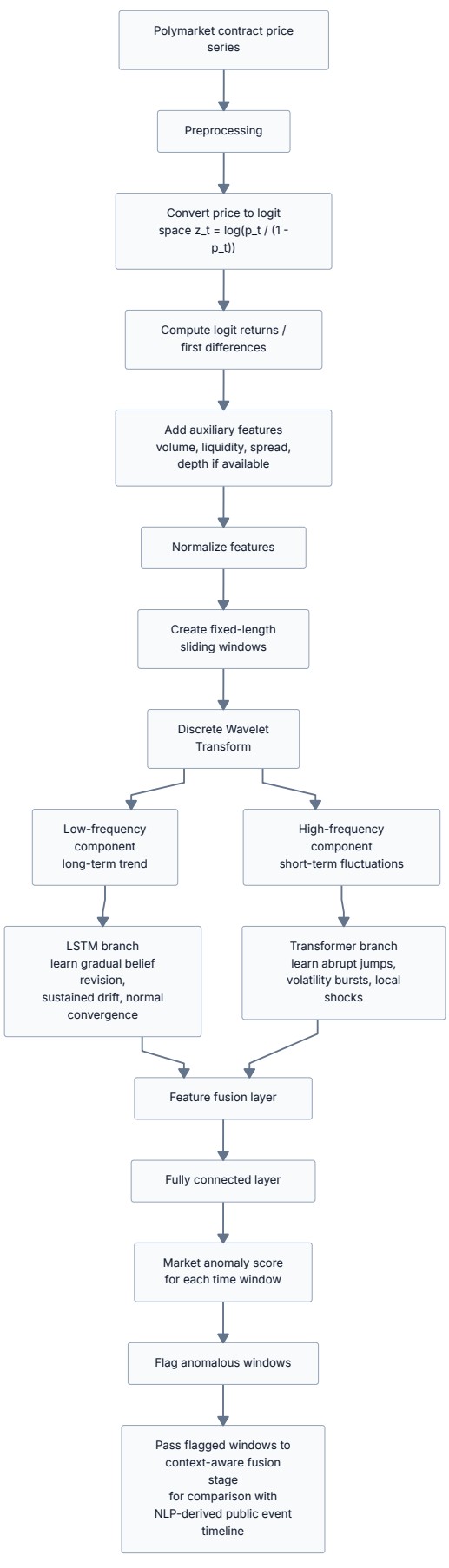

Fig. 3. WaveLST-Trans Architecture

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
