# OpenReview forum: "Towards Detecting Insider Trading in Prediction Markets: Event-Aligned Multi-Scale Anomaly Detection"
_tsinghua.edu.cn/THU/2026/Spring/ANM — THU 2026 Spring ANM Submission_

### Official Review · Reviewer_giMW · 2026-05-12

**Rating:** 8
**Confidence:** 4

**Summary:**

This proposal presents a context-aware anomaly detection framework for prediction markets. It combines multi-scale time-series anomaly detection on market prices with an NLP-based event alignment module, and outputs windows labeled as explained, partially explained, or unexplained.

**Strengths:**

The problem is timely and socially relevant. The proposal is careful not to claim direct legal proof of insider trading, and instead frames the task as anomaly triage, which is more realistic. The idea of combining price dynamics with public-event context is sensible and likely more useful than price-only anomaly detection. The project also acknowledges the validation difficulties honestly.

**Weaknesses:**

Unexplained movement does not necessarily imply insider trading. The event-alignment and scoring rules seem heuristic and may be brittle.

**Questions:**

How will you define a fair baseline for “explained vs unexplained” movement? What sources will you trust for event timestamps and why?

---

### Official Review · Reviewer_43e7 · 2026-05-15

**Rating:** 8
**Confidence:** 3

**Summary:**

This project proposes a context-aware anomaly detection architecture for prediction markets (specifically Polymarket) that combines multi-scale time series analysis with event-alignment. The system has two main components: (1) a market anomaly detection module adapted from WaveLST-Trans that uses wavelet decomposition, LSTM, and Transformer branches to produce an anomaly score; and (2) an NLP pipeline that collects and analyzes public information (news, official statements, social media) to produce a public explanation score. These are fused into a "suspiciousness score" that classifies anomalous windows as explained, partially explained, or unexplained — with unexplained pre-event movements flagged as candidates for further investigation. The paper includes a detailed mathematical formulation of the WaveLST-Trans adaptation.

**Strengths:**

The WaveLST-Trans adaptation is described with mathematical rigor (logit transformation, wavelet decomposition, LSTM/Transformer branches, feature fusion, self-supervised loss). The NLP pipeline (relevance scoring, direction scoring, novelty, credibility, timing) is also well-specified.

**Weaknesses:**

The NLP pipeline requires sophisticated capabilities (relevance determination, directional scoring, novelty assessment, credibility scoring) that are each themselves hard research problems. The proposal glosses over the difficulty of each of these sub-tasks.

**Questions:**

How do you handle prediction markets that resolve based on events that are inherently unpredictable？

---

### Official Review · Reviewer_H9oU · 2026-05-16

**Rating:** 7
**Confidence:** 4

**Summary:**

This proposal studies anomaly detection for possible insider trading or informed trading behavior in prediction markets. The authors propose an event-aligned multi-scale anomaly detection framework that combines market time-series anomaly detection with public-event context. Specifically, Polymarket price series are transformed into logit probability space, decomposed into low-frequency trend and high-frequency shock components, and modeled using a WaveLST-Trans-inspired architecture. The detected market anomalies are then aligned with public events collected from official sources, news, and social media. The final output is a ranked list of anomalous windows labeled as explained, partially explained, or unexplained. The proposal correctly emphasizes that the goal is not to legally prove insider trading, but to provide a triage mechanism for suspicious pre-event market movements.

**Strengths:**

The proposal addresses an interesting and timely problem. Prediction markets are increasingly popular, and detecting unusual pre-event price movements is a meaningful and practical task.
The proposal makes a useful distinction between statistical anomaly and contextual suspiciousness. A large price movement should not automatically be treated as suspicious if it can be explained by public information.
The multi-scale design is reasonable. Separating low-frequency trend behavior from high-frequency shocks matches the characteristics of prediction markets, where both gradual belief updates and sudden jumps may occur.
The framework has good interpretability. Instead of only outputting anomaly scores, it classifies flagged windows as explained, partially explained, or unexplained, which is useful for human investigation.
The overall pipeline is clearly described. The proposal includes the market anomaly detection module, the NLP-based public event module, and the final fusion stage, making the system structure easy to understand.
The task is framed carefully. The authors do not claim that the system can prove insider trading, but position it as a triage mechanism for identifying candidate windows that deserve further investigation.

**Weaknesses:**

Validation remains difficult because verified insider trading labels are very rare. Without reliable ground truth, it may be hard to prove that the system detects genuinely meaningful suspicious behavior rather than only producing plausible case studies.
The public event module may introduce subjectivity. Event collection, relevance scoring, novelty scoring, and credibility scoring are central to the method, but the proposal does not fully explain how these scores will be computed or validated.
The NLP component may be too ambitious for the project timeline. The proposal includes contract understanding, relevance matching, directional scoring, novelty scoring, and credibility scoring, all of which require careful implementation.
The baseline plan is not specific enough. The proposal mentions comparing price-only anomaly detection with event-aligned anomaly detection, but it would be stronger to include concrete baselines such as rolling z-score, CUSUM, ARIMA residuals, Isolation Forest, or simple event-study-style methods.
The use of synthetic data needs more detail. The proposal mentions using generative tools or LLMs to create synthetic evaluation data, but it is unclear how realistic the synthetic suspicious patterns will be and how the authors will avoid making the evaluation too aligned with their own assumptions.
Even if a price movement is not explained by available public information, it still does not necessarily indicate insider trading. The final system should clearly communicate this uncertainty to avoid overinterpretation.
Many prediction markets may have sparse trading, low liquidity, or noisy price movements. The proposal should specify minimum liquidity filters and explain how markets with limited volume or missing data will be handled.

**Questions:**

How will you define and validate the labels “explained,” “partially explained,” and “unexplained”?
What specific baselines will be used for comparison? Will you include simple statistical detectors such as rolling z-score, CUSUM, ARIMA residuals, or Isolation Forest?
How will the public explanation score be computed in practice? Is it rule-based, LLM-based, manually labeled, or trained from data?
How will you evaluate the NLP event-alignment module separately from the time-series anomaly detector?
What criteria will you use to select Polymarket markets? Will there be minimum requirements for trading volume, liquidity, or event clarity?
How will synthetic data be generated, and how will you ensure that synthetic suspicious patterns are realistic rather than too easy for the proposed method?
How will the system handle cases where public information exists but is delayed, ambiguous, multilingual, or spread through informal social media channels?
How will you prevent the final ranking from overemphasizing pre-event movements that are actually caused by unobserved but public information?

---

### Official Review · Reviewer_FzZX · 2026-05-16

**Rating:** 7
**Confidence:** 4

**Summary:**

This proposal studies anomaly detection for prediction markets, especially detecting suspicious pre-event price movements that may indicate informed trading. The proposed system combines a multi-scale price anomaly detector with an event-context layer that checks whether public information can explain the movement.

**Strengths:**

The scope is appropriately cautious. The proposal does not claim to prove insider trading; instead, it frames the system as an anomaly-triage tool for surfacing suspicious windows. This is a good and defensible framing.

The methodology is concrete. It defines logit transformation, belief updates, wavelet decomposition, LSTM/Transformer branches, anomaly scoring, and rule-guided fusion with public event evidence.

**Weaknesses:**

The validation plan is weak. Since verified insider-trading labels are rare, the proposal should define more precise evaluation criteria, such as whether the event-context layer reduces false positives, whether known public events are correctly matched, and whether synthetic pre-event anomalies are recovered.

---

### Official Review · Reviewer_nyKc · 2026-05-16

**Rating:** 7
**Confidence:** 4

**Summary:**

This proposal presents a lightweight, context-aware anomaly detection framework for prediction markets such as Polymarket, designed to identify potential insider trading. The proposed method first transforms prediction market prices into the logit probability space and applies an adaptive WaveLST-Trans model for multi-scale time series analysis. Specifically, the model uses wavelet decomposition to separate low-frequency trends (modeled by an Long Short-Term Memory network) from high-frequency abrupt changes (modeled by a Transformer), thereby generating a market anomaly score. The system then integrates this score with public event context extracted through an Natural Language Processing pipeline. Finally, the detected anomalous windows are categorized into three classes: explained, partially explained, and unexplained, where unexplained price movements are treated as candidate anomalous trades requiring further investigation.

**Strengths:**

1. The proposal innovatively introduces finacial anomaly detection methods to prediction market and employs logit space to process probability time series.
2. By introducing public information, the design will hopeful achieve effective distinguishment between Normal news-driven repricing and anomalies.

**Weaknesses:**

1. I hope the authors could give a more detailed explanation of how "prediction market" operates. It is still a little hard for those who has never known "prediction market" to understand some parts of this proposal.
2. Since there are not enough valid and convincing labels, it's hard for the method to undergo rigorous supervised evaluation.
3. To the best of my knowledge, WaveLST-Trans can't give any more information about the anomaly type beyond anomaly score. And as a reconstruction/forecasting based anomaly detection method, it cannot be straightforwardly extended to output specific anomaly types. The realizability of this part is suspectable.

**Questions:**

1. Could the authors offer a little more information about your plan about how to get the anomaly type?

---

### Official Review · Reviewer_mT1Z · 2026-05-16

**Rating:** 8
**Confidence:** 3

**Summary:**

This proposal introduces a lightweight, context-aware anomaly detection framework designed to identify potential insider or informed trading within prediction markets like Polymarket. The primary objective is to detect suspicious, pre-event price movements and provide a reliable triage mechanism for further investigation, rather than establishing legal proof of insider trading.  By aligning the detected market anomalies with the public event data, the framework evaluates whether external information explains the trading behavior. The final output of the system is a ranked list of anomalous windows categorized into three distinct classes: explained, partially explained, and unexplained. Price movements categorized as "unexplained" serve as the primary candidate anomalous trades flagged for deeper analysis.

**Strengths:**

1. The proposal tries to solve a real world problem that become rampant in recent months as predict markets are growing fastly and attracted public attention.

2. The description of the proposed methodology is clear and detailed.

3. The observation that the insider trading detection is actually some anomaly-triage is inspiring and interesting.

**Weaknesses:**

As there are only few verified insider-trading events in the real world, the final evaluation plan could be week if synthetic data by LLM consist a substantial part of the final test set.

---

### Official Review · Reviewer_WLGr · 2026-05-17

**Rating:** 10
**Confidence:** 4

**Summary:**

This architecture uses WaveLST-Trans to separate Polymarket price data into low-frequency trends and high-frequency shocks. It matches these market anomalies with news events to triage trades as "explained" by public info or "unexplained" (suspicious). The final goal is to create a ranked list of investigative leads for potential insider trading.

**Strengths:**

The logit-space transformation is a brilliant domain-specific adjustment for bounded probability data. The "context-aware" triage effectively reduces false positives by ignoring market jumps that happen after news is already public.

**Weaknesses:**

Low-liquidity markets in the prediction space often produce "noisy" price ticks that can look like anomalies but are actually just artifacts. There is a severe lack of verified real-world labels for insider trading, meaning success is mostly measured through synthetic data.

---

### Official Review · Reviewer_AuSu · 2026-05-17

**Rating:** 8
**Confidence:** 4

**Summary:**

This proposal introduces an event-aligned anomaly detection architecture designed to identify potential insider trading in prediction markets like Polymarket. The system combines a time-series model (an adapted WaveLST-Trans) to detect statistical anomalies in market probabilities with an NLP pipeline that processes public information. By cross-referencing market shocks with public news timelines, the framework categorizes market movements as explained by public data, partially explained, or unexplained (flagged as suspicious).

**Strengths:**

- The motivation is highly timely and relevant. With the recent explosion of prediction markets, identifying informed manipulation versus organic news reactions is a pressing real-world challenge.
-The architectural separation of "statistical unusualness" (the market anomaly score) from "contextual suspiciousness" (the NLP-derived explanation score) is a highly practical and intelligent design choice for triage.
- Transforming binary prediction market prices into logit space before calculating belief updates is a mathematically sound approach that properly accounts for the bounded nature of probability.
- The use of wavelet decomposition to route long-term probability drift to an LSTM and abrupt shocks to a Transformer is a very fitting application of the WaveLST-Trans model to this specific domain.

**Weaknesses:**

- The NLP pipeline described (extracting relevance, direction, novelty, and credibility from unstructured text) is incredibly ambitious and essentially a massive research project on its own. The proposal glosses over exactly how these subjective metrics will be reliably quantified (e.g., are you using zero-shot prompting with LLMs, or training custom classifiers?).
- Validation strategy: The authors correctly identify the lack of ground-truth insider trading labels as a major challenge, but relying on LLM-generated synthetic data for evaluation introduces heavy bias. If an LLM generates the synthetic insider trading, and an LLM is used in the NLP pipeline to detect it, the evaluation might just measure the LLM's self-consistency rather than real-world efficacy.
- Low liquidity is mentioned as a challenge, but prediction market order books are often so thin that a single $50 trade can cause a massive price spike. The proposed liquidity confidence score feels like an afterthought compared to the deep learning components.

**Questions:**

- What specific models or techniques will power the NLP pipeline to accurately score "novelty" and "credibility" in real-time?
- How will you ensure the LLM-generated synthetic dataset accurately reflects the complex microstructure and order-book dynamics of real insider trading?
- Will the WaveLST-Trans model be trained globally across all Polymarket contracts, or fine-tuned per contract? (The latter might be impossible for short-lived events).

---

### Official Review · Reviewer_W2zp · 2026-05-18

**Rating:** 8
**Confidence:** 3

**Summary:**

The authors propose a context-aware anomaly detection framework for prediction markets (Polymarket), aimed at flagging potential insider trading cases. The system has two parallel components: (1) price-side anomaly detector adapted from WaveLST-Trans (wavelet decomposition into low-high frequency with LSTM for slow trends, Transformer for sudden shocks), and (2) an NLP pipeline that collects and scores public information from news, official statements, and social media. Then,a fusion layer combines the market anomaly score with a public explanation score and a pre-event timing score, producing a final label: explained, partially explained, unexplained, or low-liquidity artifact.

**Strengths:**

- The framing is well-scoped. The authors position the system as a triage tool rather than a legal classifier, which is the right level of ambition for the problem.
- The distinction between statistical unusualness and contextual suspiciousness is a real conceptual contribution. Operationalizing it through a pre-event timing score is intuitive and well-motivated (movements before public news matter more).
The mathematical appendix is well-written. Logit for bounded probability data, wavelet decomposition, the LSTM/Transformer split, and the Huber + reconstruction loss are all specified enough precisely to implement.
The pipeline is modular: market-side and event-side analyses are decoupled and combined only at the fusion stage, making the system easy to modify and correct.

**Weaknesses:**

- The NLP pipeline (relevance, direction, novelty, credibility scoring) is ambitious. Each sub-task is itself a small research problem. A simpler fallback manually curated event table for the first iteration, such as from automated scraping would make the timeline more realistic.
Baselines are underspecified. The proposal mentions comparing price-only against event-aligned versions, but standard anomaly detection baselines (rolling z-score, CUSUM, ARIMA residuals, Isolation Forest) would strengthen the evaluation.
Validation without verified ground truth is the hardest part. Synthetic data from LLMs risks circularity. Two or three qualitative case studies on documented cases (e.g., the Maduro arrest bet) would probably be more convincing than aggregate metrics on synthetic data.

**Questions:**

How will the public explanation score be computed in practice?
How will the four fusion thresholds mentioned (τ_M, τ_E, τ_P, τ_Q) be calibrated without having labeled data?
Do you have specific historical cases or events in mind for qualitative validation?

---

### Official Review · Reviewer_uvG2 · 2026-05-18

**Rating:** 6
**Confidence:** 4

**Summary:**

[AI Review] This paper proposes an event-aligned multi-scale anomaly detection system for identifying insider trading in prediction markets. The core idea combines price-based anomaly detection with NLP-driven event context alignment. While the problem framing is clear and the writing quality is high, the proposal suffers from critical gaps: the NLP pipeline is undefined vaporware with no algorithmic specification, there is no evaluation protocol (no metrics, baselines, or ground truth), and the proposed fusion mechanism is a hand-tuned linear combination rather than a learned model. The novelty claims are overstated relative to decades of event-study methodology in finance.

**Strengths:**

1. Excellent writing quality and organization (8/10), making the proposal easy to follow.
2. Commendable honesty about limitations, which is rare in project proposals.
3. Strong mathematical appendix (Eqs. 5-23) providing formal grounding.
4. Clear problem framing that distinguishes statistical unusualness from contextual suspiciousness.
5. Reasonable narrowed scope appropriate for a class project timeline.

**Weaknesses:**

1. The NLP pipeline for event alignment—the core novelty claim—is completely undefined: relevance, direction, novelty, and credibility scoring have zero algorithmic specification, and Eq. (1) is pseudocode with no model, API, or prompt named.
2. No evaluation protocol exists: no metrics, no baselines, no ground truth methodology, and synthetic data generation is unspecified; 'improves interpretability' is not an evaluable claim.
3. The fusion mechanism (Eq. 2: S_t = α·M_t + β·P_t - γ·E_t) is a hand-tuned linear combination, and labeling rules (Eq. 4) use hardcoded thresholds—this is two independent pipelines connected by a spreadsheet formula, not learned end-to-end.
4. Novelty is overstated: event study methodology has combined price and event context for 50+ years, and the claim that 'no prior work integrates external market context' is false.
5. The title is misleading—the system flags unexplained price movements, not insider trading specifically.
6. Foundation risk from WaveLST-Trans reference [6] published in a low-visibility venue.
7. Unrealistic 4-week timeline when 12+ weeks would be needed.
8. Logit transform singularity at p=0 and p=1 is unaddressed.
9. Liquidity confidence score Q_t is not integrated into the composite score S_t.
10. 12+ free hyperparameters with no tuning guidance provided.

**Questions:**

1. Can you provide concrete algorithmic specification for the NLP event-alignment module, including which models, APIs, or prompts would be used?
2. What specific metrics, baselines, and ground truth methodology do you plan to use for evaluation?
3. Why was a hand-tuned linear combination chosen over a learned fusion approach, and have you considered end-to-end training?
4. How do you address the logit transform singularity at p=0 and p=1?
5. Why is the liquidity confidence score Q_t not integrated into the final composite anomaly score S_t?
6. What hyperparameter tuning strategy do you intend to use given the 12+ free parameters?
7. Would you consider softening the novelty claims and changing the title to more accurately reflect the system's scope?